# Performance Improvement of Glass Microfiber Based Thermal Transpiration Pump Using TPMS

**DOI:** 10.3390/mi13101632

**Published:** 2022-09-29

**Authors:** Pitipat Parittothok, Chanon Poolwech, Tanawit Tanteng, Jakrapop Wongwiwat

**Affiliations:** Department of Mechanical Engineering, King Mongkut’s University of Technology Thonburi, Bangkok 10140, Thailand

**Keywords:** thermal transpiration pump, Knudsen pump, glass microfiber, triply periodic minimal surface, thermal creep flow

## Abstract

The Knudsen pump, known as a thermal transpiration membrane, is an air inducer that has been mostly studied for small-scale power generation devices. It is a porous medium that does not require any mechanically moving component, but rather uses the temperature gradient across two surfaces of the membrane to induce air from the colder side to the hotter side. If the temperature on the colder side of the membrane is reduced by a thermal guard, the pumping performance of the membrane seems to be improved. Therefore, the membrane integrating with TPMS structures as thermal guards for both experiment and simulation were conducted in this study. The results of flow rate and temperature distribution on the membrane surface were compared. Three characteristic parameters of the membrane, i.e., area factor, pore radius and permeability, were found and can be used in an equation to estimate the air flow rate through the membrane. Diamond was found to be the highest flow improvement while Primitive was the lowest flow improvement. The simulation results with varying %RD also supported that the contact area between the TPMS structure and the membrane inlet surface made Diamond conduct more heat out from the membrane surface than other TPMS structures.

## 1. Introduction

Micro-electromechanical systems (MEMS) have become more interesting to various research groups over the past few decades. Micropumps are important components for many microscale applications such as micro gas chromatographs [1,2], small satellite propulsion [3], heat pumps [4] and SOFC power generators [5]. In general, pumps or compressors use mechanical work to drive fluid from a low pressure side to a high pressure side, but at a microscale, they have to operate at extremely high speed to generate enough flow [6,7] while consuming a lot of power. One promising option for microscale devices is to use a thermal transpiration membrane that operates as a thermal transpiration pump known as a Knudsen pump. If the temperature on one side of the membrane is higher than on another side, air will be induced from the lower temperature side to the higher temperature side. This phenomenon was used to induce gas into microscale devices from the temperature difference across the membrane.

Thermal transpiration is a rarefied gas dynamics phenomenon describing the gas flow through a narrow channel with an imposed temperature gradient. Various porous materials have been explored to perform as thermal transpiration pumps since 1879 by Osborne Reynolds using biscuit ware, stucco and meerschaum [8]. He found slight pressure generated from these porous materials after applying steam on one side and water on another side of porous materials. Other porous materials were also investigated recently such as aerogel [9], zeolite [10], cellulose ester [11] and nanoporous ceramic [12]. All of these types of porous materials need to have small pore sizes and low thermal conductivity to increase temperature differences across the materials themselves. Another promising thermal transpiration membrane material that has been also investigated for several years was borosilicate glass filters [13,14,15]. They demonstrated that borosilicate glass filters could be used to induce gas and operate as compressors.

Many research groups have developed numerical models to estimate net flow that depends on the Knudsen number and flow coefficient [16,17]. Other aspects of the thermal transpiration pump have also been studied, such as different gases and multi-stage pumps [18,19,20] and modified governing equations of net flow were introduced and compared with experiments. More recently, flow characteristics and pumping mechanisms were studied by computational fluid dynamics [21,22] and different gases were studied using Direct Simulation Monte Carlo [23,24]. The governing equation of gas flow from thermal transpiration pump was developed by each research group and ended up with slightly different forms. It seems to be inconclusive for different studies and different apparatuses. However, the common term that affects flow rate the most is the temperature difference across the membrane. An idea of using a thermal guard [15] or a thermal management platform [25] with a thermal transpiration membrane was introduced and enhanced flow rate by uniforming and decreasing the inlet surface temperature. Fins on the inlet side of the membrane might be a key component to reduce inlet temperature and improve thermal transpiration performance.

Applying conventional fins on the inlet side of the membrane might not improve thermal transpiration performance, because these fins seem to block the inlet flow. Porous metal foam with high thermal conductivity was studied for heat transfer purposes [26]; however, the non-uniform structure of the metal foam is difficult to predict and replicate heat transfer performance. There are still some repeated structures, known as lattice structures [27,28] and Triply-Periodic Minimal Surface (TPMS) that is on hyperbolic plane in three-dimensional space with zero mean curvature [29,30]. TPMS had been fabricated to study their mechanical properties including the simplification of TPMS models with an equivalent entity [31]. For strut-based lattice structures, they were used for heat transfer purposes [32,33,34,35] and the results showed that strut-based lattice structures have a higher heat transfer coefficient than conventional fins. However, strut-based lattice structures have a high-stress concentration at strut junctions [36], while TPMS structures provide smooth and continuous surfaces. Moreover, the heat transfer coefficient and effective thermal conductivity of TPMS structures are also significantly higher than strut-based lattice structures [37,38], because of their continuous surface patterns. The study of heat transfer from pin-fin heat sinks demonstrated that thin-walled structures of TPMS lattices had high performance in heat dissipation [39]. Different shapes of TPMS structures were studied to be used inside heat exchangers [40,41] and as a cooling component for injection molds [42]; however, there are still other shapes of TPMS structures [43] that might be suitable for thermal transpiration membrane as depicted in Table 1.

The performance of the thermal transpiration pump depends on the temperature gradient across the membrane. If the temperature on the inlet side of the membrane is decreased, more air will be induced through the membrane. Heat distributed plates have been investigated in some previous studies, but none of them have been studied combining with TPMS structures that potentially improve the pumping performance of the membrane. Therefore, the thermal transpiration membrane combined with structures of TPMS placed on the inlet side of the membrane was investigated in this study to improve thermal transpiration pumping performance. The experimental study was investigated as well as the modeling of thermal transpiration membrane in ANSYS-Fluent for microscale applications.

## 2. The Modeling of Thermal Transpiration Membrane

Thermal transpiration flow occurs when gas molecules collide with high-temperature walls and the kinetic energy of molecules is increased because they gain heat from the walls. Some gas molecules bounce in the colder direction and some gas molecules bounce in the hotter direction; however, the net flow of gas goes towards the higher temperature side. This phenomenon is called thermal creep flow which can be expressed by the theory of rarefied gas dynamics through a micro-scale channel. The channel has to be as small as the scale of the mean free path of a gas molecule. In a narrow channel, thermal creep flow usually occurs near the channel wall. When one side of the channel has higher pressure than another side, Poiseuille flow which is a pressure-induced flow also occurs in the region far from the wall as depicted in Figure 1.

In this current work, the air flow rate generated by thermal transpiration membrane is based on the equation described as
(1)$$\dot{M}=P_{avg}\sqrt{\frac{m}{2kT_{avg}}}A\frac{L_{r}}{L_{x}}\left[\frac{∆T}{T_{avg}}Q_{T}-\frac{∆P}{P_{avg}}Q_{P}\right]$$
where thermally driven flow coefficient ($Q_{T}$) and pressure-driven return flow coefficient ($Q_{P}$) were originally taken from linearized Boltzmann equation and Knudsen Number (Kn) which is the ratio of gas mean free path to channel radius. All other parameters were explained in the previous work [18]. However, this work considered a thermal transpiration pump from a perfect cylinder that did not occur in borosilicate glass filters. The previous study has shown that the equation of mass flow rate had to include the effect of backflow through porous media in the equation to predict air flow rate through the thermal transpiration membrane [13]. The backflow in the study is expressed by Darcy’s law which describes flow through porous media depending on the permeability of the media, viscosity of the fluid, the distance and the pressure drop across the media. Therefore, the modified mass flow rate can be rewritten as
(2)$$\dot{M}=P_{avg}\sqrt{\frac{m}{2kT_{avg}}}A_{tot}\left(AF_{TT}\right)\frac{L_{r}}{L_{x,tot}}\left[\frac{∆T}{T_{avg}}Q_{T}-\frac{∆P}{P_{avg}}Q_{P}\right]-\rho_{avg}\frac{\kappa \left(A_{tot}\left(1-AF_{TT}\right)\right)\Delta P}{\mu L_{x}}$$
and volume flow rate can also be considered as
(3)$$\dot{V}=\frac{P_{avg}}{\rho_{avg}}\sqrt{\frac{m}{2kT_{avg}}}A_{tot}\left(AF_{TT}\right)\frac{L_{r}}{L_{x,tot}}\left[\frac{∆T}{T_{avg}}Q_{T}-\frac{∆P}{P_{avg}}Q_{P}\right]-\frac{\kappa \left(A_{tot}\left(1-AF_{TT}\right)\right)\Delta P}{\mu L_{x}}$$

A concept of airflow through a borosilicate glass filter is depicted in Figure 2. The microscopic view of the filter shows random bamboo-like structures, not a perfect cylinder as in the assumption of Equation (1); therefore, some areas of the entire membrane should behave as thermal transpiration membranes. The first term in Equation (3) is thermal transpiration flow from temperature gradient across the membrane that originally depends on flow coefficients ($Q_{T}$ and $Q_{P}$) and Knudsen Number. The second term in Equation (3) is backflow through porous media from Darcy’s law in the region that does not correspond to the flow from thermal transpiration.

## 3. Experimental Setup

### 3.1. Apparatus

The pumping performance and all related parameters of the thermal transpiration membrane were measured by the apparatus expressed in Figure 3. Firstly, a glass microfiber filter grade GF/F membrane, manufactured by Cytiva, United Kingdom, was clamped between the chamber flange and the membrane holder. A 200 W cartridge heater was placed inside the chamber as a heat source on the outlet side of the thermal transpiration membrane. Type K thermocouples were carefully placed on both the hot side and the cold side of the membrane to record temperature on surfaces. The outlet of the chamber was connected to a bubble flow meter and Sensirion SDP-1000L differential pressure sensor to measure air flow rate and pressure generated by the membrane. The temperature and pressure of this setup were recorded by NI-cDAQ 9174 datalogger. A pressure regulator at the outlet of the test section could be varied to control the pressure difference across the membrane while the flow rate was measured by a bubble flow meter.

The experimental study on the performance improvement of thermal transpiration membrane was firstly conducted only on the membrane without TPMS structure to identify area factor (AF), pore radius ($L_{r}$) and permeability (κ) as constant parameters of the membrane. Then each of the six different TPMS structures was carefully placed on the inlet side of the membrane to lower the surface temperature of the membrane and increase the airflow rate. The experiment was also repeated with different TPMS structures and temperature, pressure and airflow rate were also recorded to compare with simulation results in the following sections.

### 3.2. Boundary Conditions of Thermal Transpiration Membrane Simulation

An aluminum case, a heater, a thermal transpiration membrane and TPMS geometries from the experimental setup were created by CAD software and imported to ANSYS-Fluent 2021 R2 to study all related parameters as depicted in Figure 4 for both with and without TPMS geometries on the membrane inlet surface. The inlet pressure on the inlet side was set to 0 bar gauge and the outlet pressure was varied to obtain performance curves of the thermal transpiration membrane that will be explained in the following section. The turbulence model was not included in the simulation, because the Reynolds number was below 1 for all conditions. Surface-to-Surface Radiation Model, a built-in radiation model in ANSYS-Fluent, was selected to estimate radiation in the simulation. The flow of air in the simulation was only driven by a membrane with integrated User-Defined Functions (UDFs).

The modeling and equations of flow through thermal transpiration membrane varied from different research groups as mentioned previously. Therefore, Equation (2) has been chosen in this study to be implemented in UDFs to calculate the mass flow rate of air through the membrane including the effect of thermal creep flow, Poiseuille flow and Darcy flow altogether in a single equation. In the simulation, the membrane has to be able to prevent heat from transferring out from the chamber while driving or pumping air from the colder side to the hotter side of the membrane. The membrane structure was divided into three layers to involve both effects of heat transfer and air pumping as demonstrated in Figure 5 which was developed first time in this study to be able to work smoothly with ANSYS-Fluent. By splitting the membrane into 3 layers, the middle layer was defined as a solid layer to fulfill the material property of the membrane, both two outer layers were fluid layers to define the pumping mechanism in the simulation model by giving one side of the membrane to consume air as a mass sink and another side to generate air as a mass source. Equation (2) was implemented in UDFs by finding the temperature and pressure data from both sides of the mass sink and mass source layers to calculate the airflow rate. Finally, the airflow rate at the outlet side was set to be equal to the airflow rate at the inlet side.

### 3.3. TPMS Structures and Fabrication

All 6 TPMS structures, created by nTopology using equations given in Table 1, could be described by well-known variables such as the ratio of solid material to the volume of a TPMS unit cell (%RD), the unit cell length (L), the biggest sphere that can fit inside 1 unit cell of TPMS shape ($D_{pore}$) and mean thickness of sheet based TPMS structure (t). Parameters that defined geometries of TPMS are shown in Figure 6 and an example of printed TPMS structure is shown in Figure 7. TPMS structures were created by the Curve generation feature in nTopology for a precise design. X, Y and Z positions for unit cell size, and t for the thickness could be modified to fit TPMS structures with the apparatus and simulation. After finishing with setting up all TPMS parameters, nTopology could export all geometries for 3D printing and also transfer them to ANSYS-Fluent for simulation.

## 4. Results and Discussion

### 4.1. Thermal Transpiration Membrane without TPMS

The testing of the thermal transpiration membrane consisted of multiple steps. First of all, a membrane was tested repeatedly with varying heat input power at 12.1 W, 16.3 W and 23.5 W to apply average different temperature gradients across the membrane at 67.3 K, 82.6 K and 104.4 K, respectively. A pressure regulator at the outlet of the test section was adjusted to vary the pressure drop and airflow rate from the membrane. Experimental results are presented in Figure 8 by solid circles. After the experiment was performed, pressure on each side of the membrane, the temperature on each side of the membrane and volume flow rate of air were substituted in Equation (3) to obtain characteristic parameters of membrane consisting of area factor (AF), pore radius ($L_{r}$) and permeability (κ) which are 0.172, 1.65 × 10^−7^ m and 4.15 × 10^−13^ m^2^, respectively. These three characteristic parameters were also used in UDF for all simulation setups. Dash lines for each heat power input show the calculated volume flow rate from the characteristic parameters with given temperature differences and varying pressure differences across the membrane. Finally, simulations in ANSYS-Fluent with UDFs including three characteristic parameters were performed to study related factors and related results that could not be observed from the experiment. The heater in the simulation was also set to the same power generation as provided by the power supply. The outlet pressure was varied to generate pressure differences across the membrane for each data point. The last characteristic property of the membrane, fitted effective thermal conductivity ($k_{m}$), was adjusted to allow temperature gradient across the membrane to fit with experimental results. The proper value of effective thermal conductivity ($k_{m}$) was found to be 0.0023 W/mK. When all results from the experiment, simulation and calculation were compared, it showed that they agreed on an acceptable trend. All characteristic parameters of the membrane that consisted of area factor (AF), pore radius ($L_{r}$) and permeability (κ) and effective thermal conductivity ($k_{m}$) were used for further study.

### 4.2. Thermal Transpiration Membrane with TPMS

After the characteristic parameters of the membrane were found in the previous section, one of the 6 TPMS structures was carefully placed one by one on the inlet side of the membrane to transfer heat out from the surface. This experiment led to the increase in temperature gradient across the membrane and increased membrane pumping performance. Solid circles in Figure 9 show recorded data points from the experiment while triangles and dash lines are data points and connecting lines from the simulation at various pressure drop across the membrane. The simulation did not modify any input parameters, which can be concluded that all characteristic properties of the membrane that were mentioned in the previous section could be used to estimate membrane pumping performance, for both with and without TPMS structures on the membrane surface. The detail of the mesh quality of simulation was discussed in Appendix A.

From the experimental results, it might be difficult to justify which TPMS structure was better than another; therefore, only simulation results were plotted in Figure 10 to compare the pumping performance of the membrane without TPMS structure and with different TPMS structures. The simulation results show all TPMS structures helped improve the pumping performance of the membrane while Diamond could increase the maximum flow rate by 8.19% compared to the membrane without TPMS, while Primitive could only increase the maximum flow rate by 5.44%. Other shapes of TPMS could improve the flow rate between 5.44% to 8.19%.

After the simulation results were investigated, they show that TPMS could decrease the temperature of the membrane surface as shown in Figure 11 for more than 10 K. It demonstrated that TPMS structures performed as heat sinks that could lower the inlet temperature and lead to a higher temperature difference across the membrane. However, different TPMS structures could improve the pumping performance of the membrane differently. If a TPMS structure is considered a heat sink, the TPMS structure thickness might be also an important factor.

### 4.3. The Effect of TMPS Structure and Pumping Performance

To investigate the relationship between the TPMS structure and pumping performance, there were three more parameters that should be introduced. The first parameter is the TPMS-air contact area ($A_{at}$) which is the area of the TPMS surface transferring heat out to the surrounding air as the yellow highlight in Figure 12. Values of $A_{at}$ of each TPMS structure are expressed in Figure 13 that IWP has the highest $A_{at}$ while Primitive has the smallest $A_{at}$. The greater the area, the more heat is released from the TPMS structure to the surrounding air.

The second parameter is the TPMS-membrane contact area ($A_{mt}$). This parameter describes the area of the TPMS structure that touches the membrane surface demonstrated as the yellow highlight in Figure 14 for all TPMS structures. All values of $A_{mt}$ are different based on the shape of the TPMS structure at the same thickness, unit cell size and pore diameter. The exact values of $A_{mt}$ can be seen in Figure 15 showing that Diamond has the largest value of $A_{mt}$ and IWP has the smallest value of $A_{mt}$.

The last parameter considered in this study was pore diameter ($D_{pore}$). It was also considered an important parameter because a TPMS structure with large $D_{pore}$ could cause a non-uniform temperature gradient across the inlet surface of the membrane. Different $D_{pore}$ of each TMPS structure is shown in Figure 16 and temperature gradients of Diamond and Primitive are shown in Figure 17 as examples.

When TPMS-air contact area, TPMS-membrane contact area and pore diameter were compared, it can be observed that Diamond provided the maximum flow rate with the largest value of the TPMS-membrane contact area, but Diamond does not have the maximum value for the other two parameters. TPMS-membrane contact area seems to have more effect on decreasing the temperature gradient of the membrane than other parameters.

Primitive has the smallest pumping performance improvement, even if Primitive does not have the smallest value of the TPMS-membrane contact area. Primitive has the smallest value of TPMS-air contact area and the largest value of Pore Diameter which could be interpreted that Primitive does not help transfer heat to the surrounding air and distribute the heat on the membrane surface as much as other TPMS structures.

### 4.4. The Effect of Relative Density on Thermal Transpiration Membrane Performance

From both experimental results and simulation results, Diamond showed the highest performance improvement of thermal transpiration membrane among other structures. Therefore, Relative Density, represented as %RD, of Diamond was varied to study the relationship between the structure thickness and the pumping performance improvement, because the greater the %RD, the thicker the Diamond structure. %RD that controls the thickness could be modified in nTopology before adding the generated geometry into ANSYS-Fluent for simulation. However, the %RD of Diamond could be varied in a limited range. If %RD is too low, the surface of TPMS will not continuously connect in every direction. If %RD is too high, the channel of TPMS will not continuously connect in every direction instead. For these reasons, 15%RD, 25%RD and 35%RD have been chosen for the study of airflow rate from the membrane. The results from varying %RD of Diamond are shown in Figure 18 that the maximum flow rate was greater if %RD was increased. It can be concluded that the greater %RD, the greater the pumping performance of the membrane.

## 5. Discussion and Conclusions

Airflow rate through thermal transpiration membrane, glass microfiber filter grade GF/F, could be estimated by adjusting three characteristic parameters which are area factor (AF), pore radius ($L_{r}$) and permeability (κ). The area factor was 0.172 which could be interpreted that only 17.2% of the entire membrane area behaved as thermal transpiration and 82.8% behaved as Darcy flow through porous media. Pore radius ($L_{r}$) was found to be 1.65 × 10^−7^ m or 0.165 μm while this microfiber filter was originally used to filter out particles from liquid and the advertised value of Particle Retention Capacity (μm) of this membrane was 0.7 μm. Even though these values could not be directly compared, pore radius ($L_{r}$) is still in the same order of magnitude. Permeability (κ) of the membrane was the parameter that relates to Darcy backflow through porous media and this value was found to be 4.15 × 10^−13^ m^2^ to use Equation (3) to estimate the flow rate.

Simulation results also demonstrated that all characteristic parameters from the experimental study could be used for the simulation with integrated UDFs. However, the effective thermal conductivity ($k_{m}$) of the middle layer of the membrane was also modified to 0.0023 W/mK to keep the temperature gradient across the membrane close to the experiment. This value seems to be lower than a normal microfiber filter because effective thermal conductivity was applied only at the middle layer of the membrane which is 3 times thinner than the actual membrane. These four values were kept constant for the rest of the study when TPMS structures were placed on the membrane.

Experimental results and simulation results of thermal transpiration membranes with a TPMS structure on the inlet surface of the membrane demonstrated that TPMS helped reduce the surface temperature of the membrane. After the surface temperature was reduced, the temperature gradient across the membrane increased, then the air flow rate through the membrane was increased. The experiment and simulation in this study supported that thermal guards or thermal management platforms in prior works help improve the pumping performance of the thermal transpiration membrane. The membrane with Diamond on the inlet surface had the highest flow improvement by 8.19% and Primitive had the lowest flow improvement by only 5.44%. The temperature contour in the simulation also confirmed that the temperature of the membrane inlet surface was reduced because a TPMS structure was placed on the surface.

After three additional parameters of TPMS structures and the membrane, i.e., TPMS-air contact area ($A_{at}$), TPMS-membrane contact area ($A_{mt}$) and pore diameter ($D_{pore}$) were investigated, the results suggested that Diamond has the largest value of $A_{mt}$ compared to other TPMS geometries. Therefore, $A_{mt}$ seems to be an important parameter that could conduct heat out from the membrane surface which led to a higher temperature gradient across the membrane, so that the highest flow improvement occurred from Diamond on the membrane surface.

In addition, tortuosity is also an interesting parameter of TPMS structures that seems to be related to the pumping performance improvement that was mentioned in some previous studies as a curvature [44,45]. However, values of tortuosity of Diamond, IWP, Primitive and Gyroid were compared and Diamond has the largest value of tortuosity [46]. It can be stated that tortuosity is also one of the key parameters that help improve the pumping performance of the thermal transpiration membrane.

From the experimental results and simulation results in this study, it can be concluded that the most effective parameter was the contact area between the TPMS structure and the membrane. The thicker the TPMS structure, the more heat transfers out from the membrane surface which causes the increase in temperature gradient across the membrane. All TPMS structures that behave as fins could improve the pumping performance of the thermal transpiration membrane. Metal 3D-printed TPMS structure, especially for Diamond, could be used to improve the pumping performance of thermal transpiration membranes for micro-scale devices.

## Figures and Tables

**Figure 1 micromachines-13-01632-f001:**
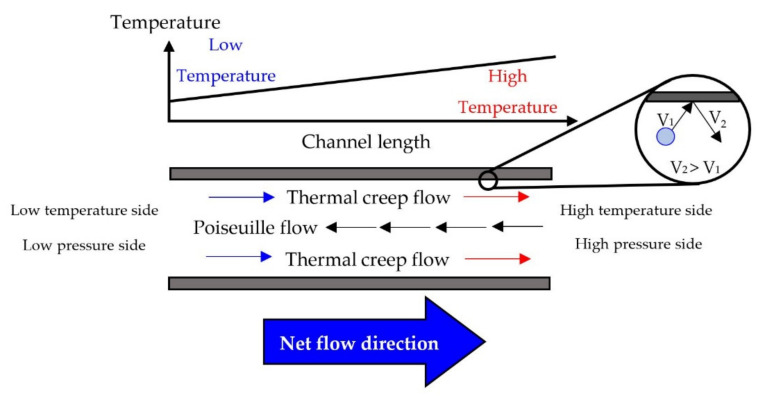
Thermal creep flow and Poiseuille flow in a narrow channel.

**Figure 2 micromachines-13-01632-f002:**
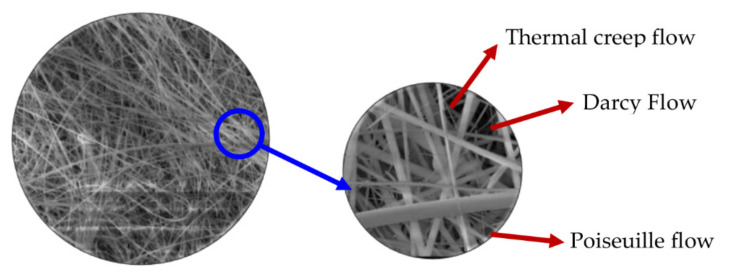
Microscopic picture of borosilicate glass filter from scanning electron microscope (SEM) (Locations of thermal creep flow, Darcy flow and Poiseuille flow are only to demonstrate that different locations have different flow behaviors.).

**Figure 3 micromachines-13-01632-f003:**
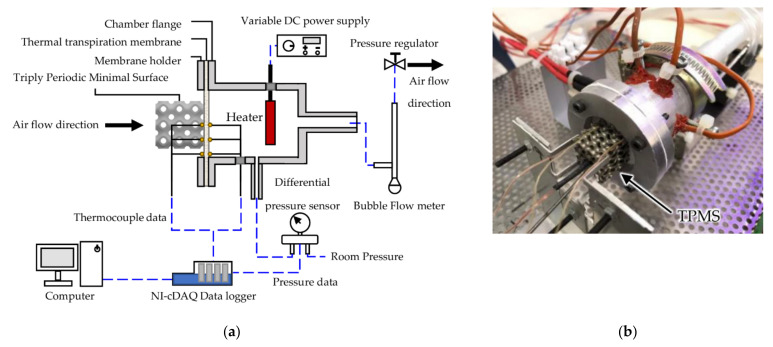
Experimental setup for membrane performance test: (**a**) schematic diagram; (**b**) as-built setup.

**Figure 4 micromachines-13-01632-f004:**
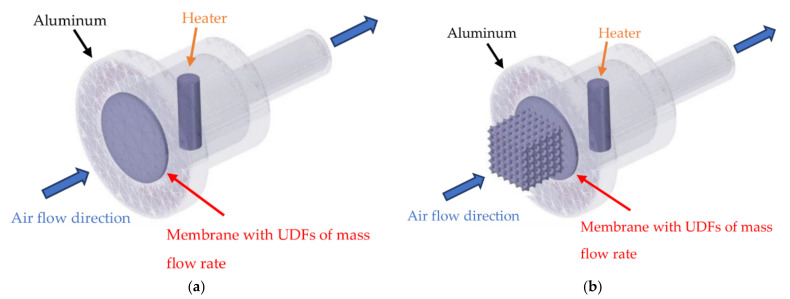
Simulation models of thermal transpiration membrane: (**a**) without TPMS geometry; (**b**) with TPMS geometries.

**Figure 5 micromachines-13-01632-f005:**
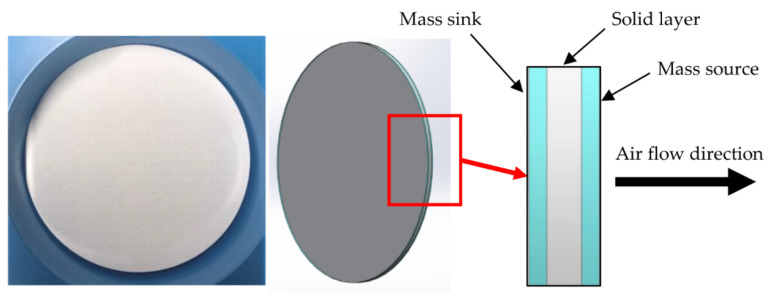
Membrane structure for the simulation consisting of a mass sink layer, solid layer and mass source layer.

**Figure 6 micromachines-13-01632-f006:**
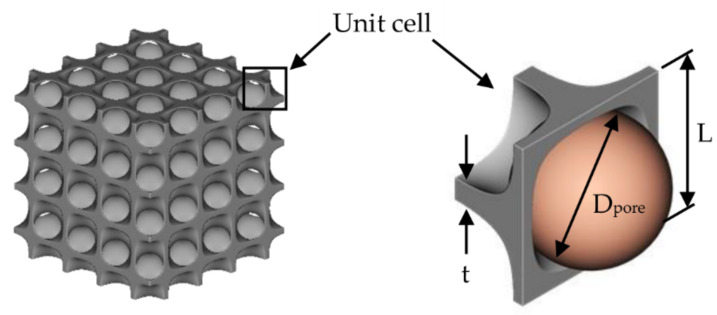
An example of parameters of TPMS generation.

**Figure 7 micromachines-13-01632-f007:**
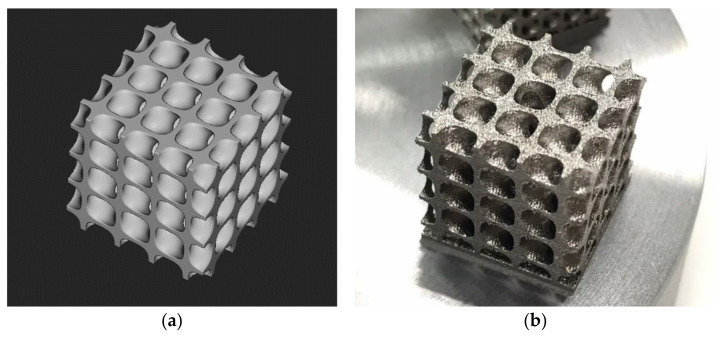
An example of TPMS structure: (**a**) generated by nTopology; (**b**) printed by Laser Powder Bed Fusion 3D printer.

**Figure 8 micromachines-13-01632-f008:**
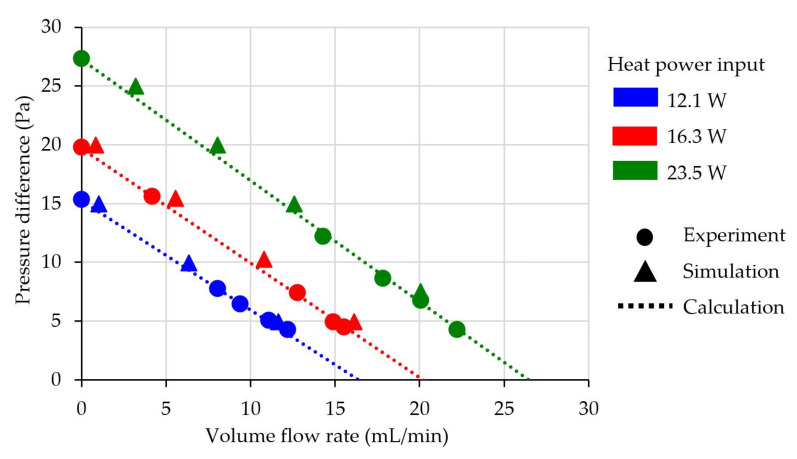
Thermal transpiration membrane performance at different heat power inputs.

**Figure 9 micromachines-13-01632-f009:**
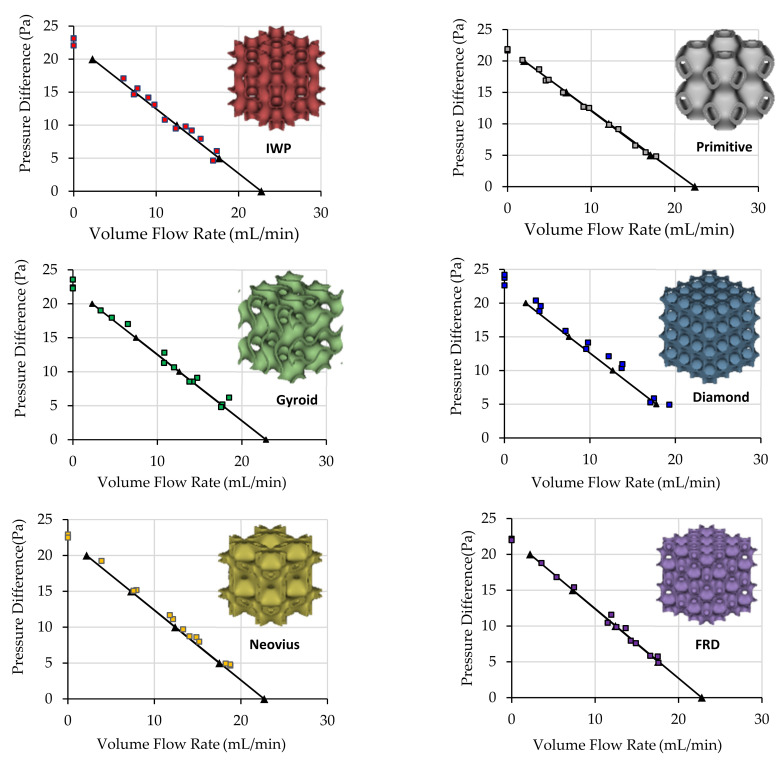
Performance curves from experimental results (rectangles) and simulation results (Triangles with a solid line) when different TPMS structures were placed on the inlet side of the membrane.

**Figure 10 micromachines-13-01632-f010:**
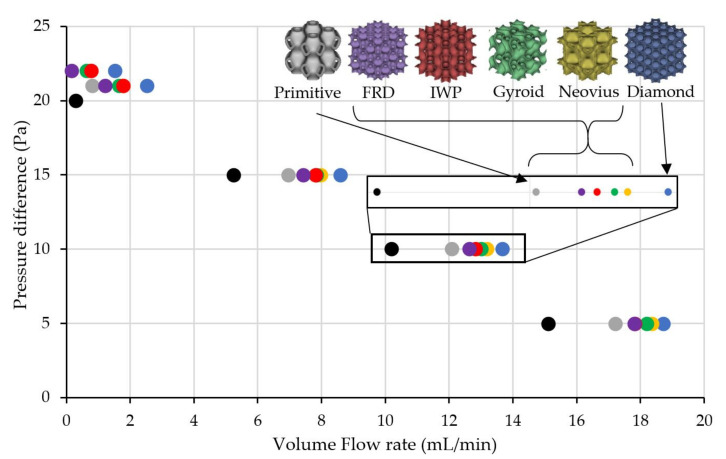
Performance curves of thermal transpiration membrane without TPMS (black circles) and with different TMPS structures (gray, purple, red, green, yellow and blue) when the power input to the heater was fixed at 16.2 W.

**Figure 11 micromachines-13-01632-f011:**
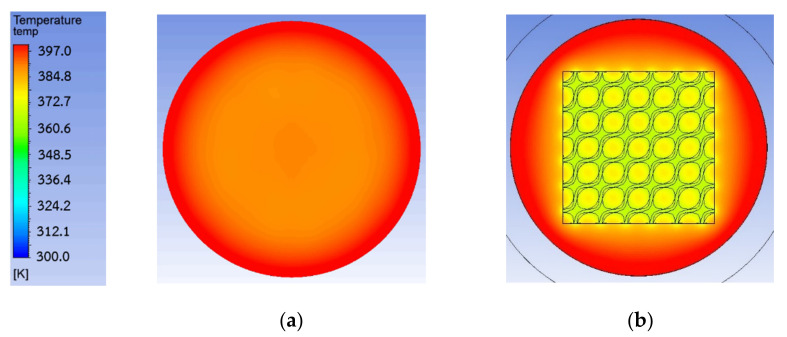
Temperature contour of membrane: (**a**) without TPMS; (**b**) with Diamond.

**Figure 12 micromachines-13-01632-f012:**
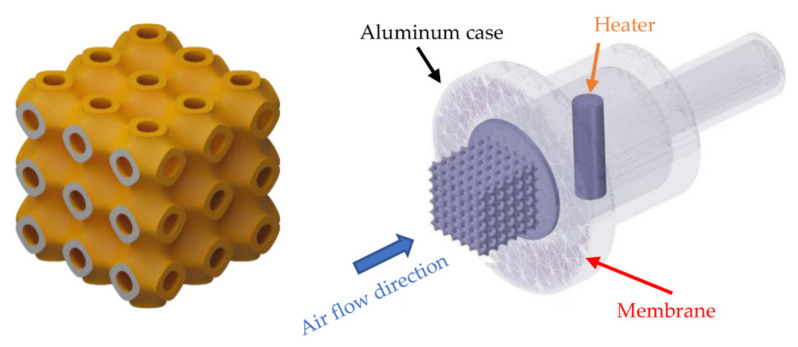
TPMS-air contact area of Primitive as an example.

**Figure 13 micromachines-13-01632-f013:**
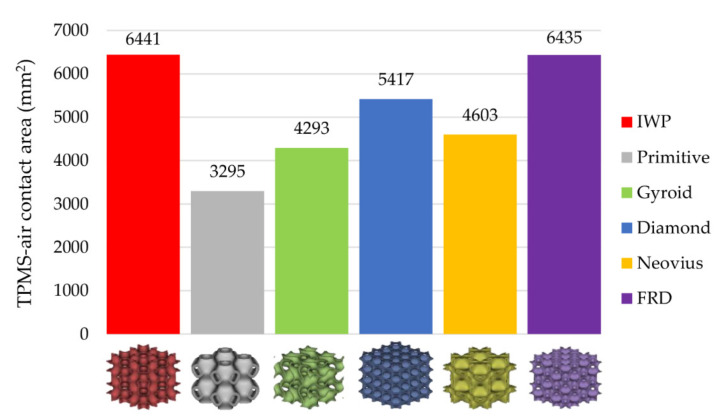
The value of the TPMS-air contact area of each TPMS.

**Figure 14 micromachines-13-01632-f014:**
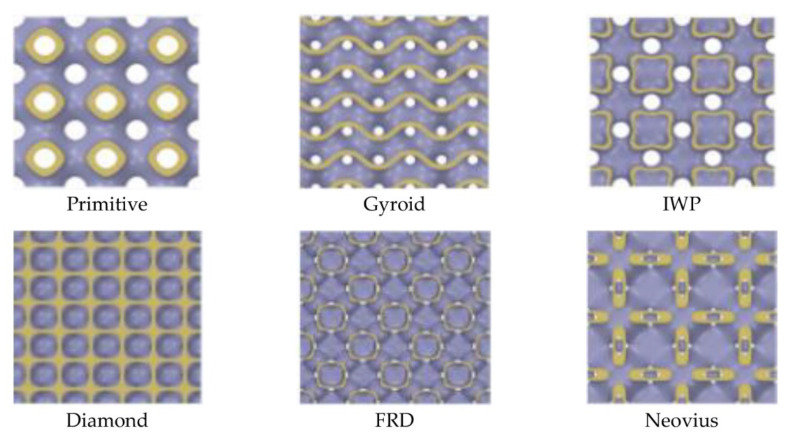
TPMS-membrane contact area of each TPMS structure.

**Figure 15 micromachines-13-01632-f015:**
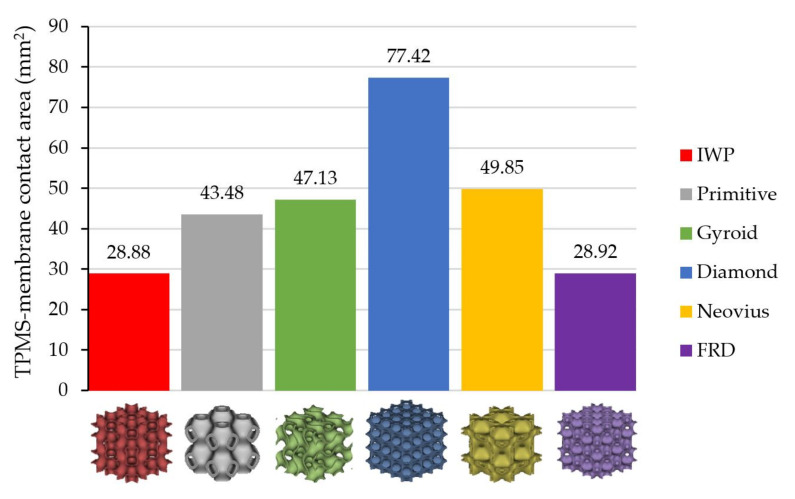
The value of the TPMS-membrane contact area of each TPMS structure.

**Figure 16 micromachines-13-01632-f016:**
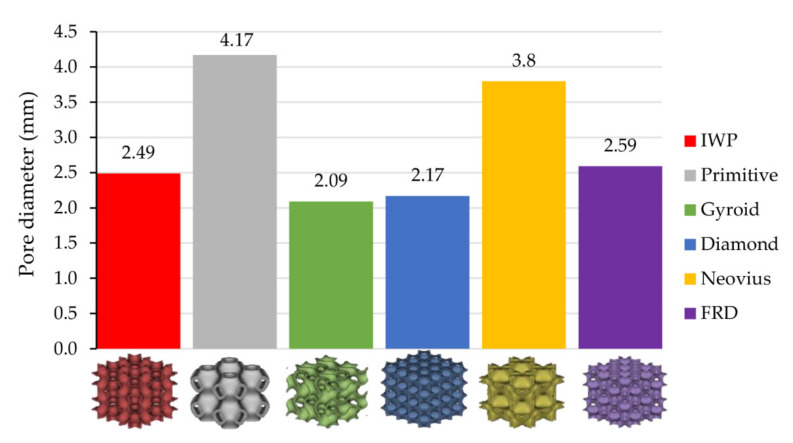
The value of pore diameter, $D_{pore}$, of each TPMS structure.

**Figure 17 micromachines-13-01632-f017:**
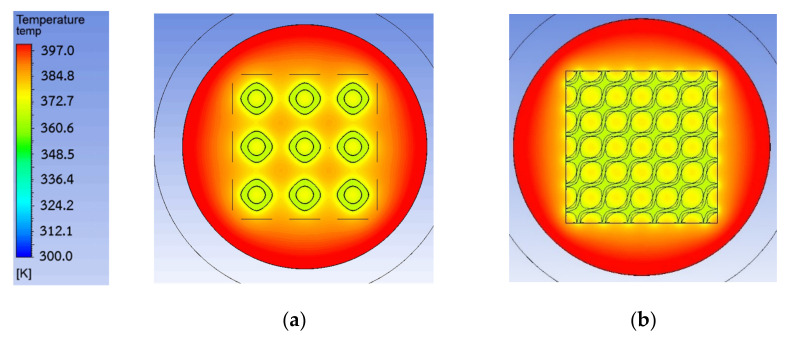
Temperature contours of membrane inlet surface between: (**a**) Primitive; (**b**) Diamond.

**Figure 18 micromachines-13-01632-f018:**
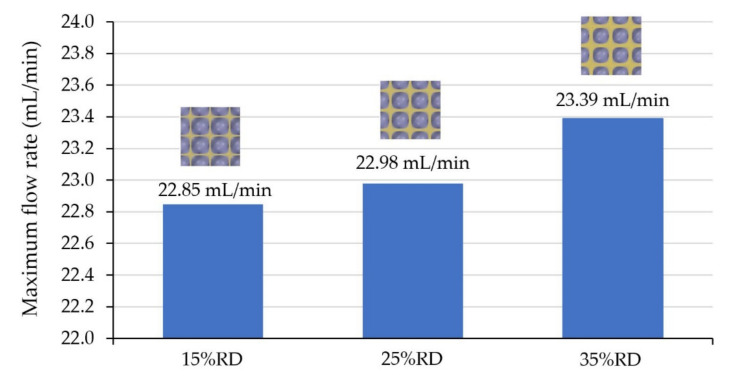
Maximum flow rate of Diamond with varying %RD.

**Table 1 micromachines-13-01632-t001:** TPMS structures and equations from a trigonometric approximation within Weierstrass Enneper formalism.

TPMS		TPMS Equation
IWP	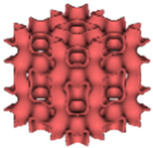	$2(\cos\left(\frac{2\pi }{L}x\right)\cos\left(\frac{2\pi }{L}y\right)+\cos\left(\frac{2\pi }{L}y\right)\cos\left(\frac{2\pi }{L}z\right)+\;\cos\left(\frac{2\pi }{L}z\right)\cos\left(\frac{2\pi }{L}x\right))-(\cos\left(\frac{2\pi }{L}2x\right)+\cos\left(\frac{2\pi }{L}2y\right)+\;\cos\left(\frac{2\pi }{L}2z\right))=c$
Primitive	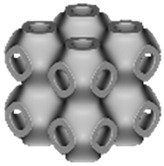	$\cos\left(\frac{2\pi }{L}x\right)+\cos\left(\frac{2\pi }{L}y\right)+\cos\left(\frac{2\pi }{L}z\right)=c$
Gyroid	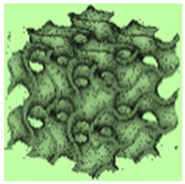	$\sin\left(\frac{2\pi }{L}x\right)\cos\left(\frac{2\pi }{L}y\right)+\sin\left(\frac{2\pi }{L}y\right)\cos\left(\frac{2\pi }{L}z\right)+\;\sin\left(\frac{2\pi }{L}z\right)\cos\left(\frac{2\pi }{L}x\right)=c$
Diamond	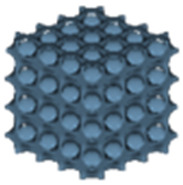	$\cos\left(\frac{2\pi }{L}x\right)\cos\left(\frac{2\pi }{L}y\right)\cos\left(\frac{2\pi }{L}z\right)-\;\sin\left(\frac{2\pi }{L}x\right)\sin\left(\frac{2\pi }{L}y\right)\sin\left(\frac{2\pi }{L}z\right)=c$
Neovius	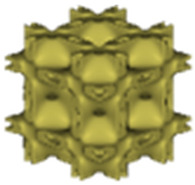	$3\left(\cos\left(\frac{2\pi }{L}x\right)+\cos\left(\frac{2\pi }{L}y\right)+\cos\left(\frac{2\pi }{L}z\right)\right)+\;4\left(\cos\left(\frac{2\pi }{L}x\right)\cos\left(\frac{2\pi }{L}y\right)\cos\left(\frac{2\pi }{L}z\right)\right)=c$
FRD	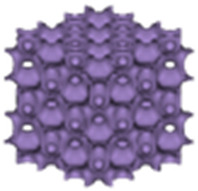	$4\left(\cos\left(\frac{2\pi }{L}x\right)\cos\left(\frac{2\pi }{L}y\right)\cos\left(\frac{2\pi }{L}z\right)\right)-\;(\cos\left(\frac{2\pi }{L}2x\right)\cos\left(\frac{2\pi }{L}2y\right)+\cos\left(\frac{2\pi }{L}2y\right)\cos\left(\frac{2\pi }{L}2z\right)+\;\cos\left(\frac{2\pi }{L}2z\right)\cos\left(\frac{2\pi }{L}2x\right))=c$

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
