# Peer review of "Performance Improvement of Glass Microfiber Based Thermal Transpiration Pump Using TPMS"

_micromachines, 2022, doi:10.3390/mi13101632_

Round 1

Reviewer 1 Report

1.      Authors are unable to identify the research gap and novelty in the study.

2.      Literature review is not well written. It includes irrelevant and mostly old references. Rewrite the introduction section by including the latest research carried out in last 3-5years. Also define and properly mention the research gap in last paragraph of the introduction section.

3.      No information is provided related to the mesh quality. Also, mesh optimization should be included in the study

4.      How are results validated?

5.      Results are not properly defined. Author should explain the result in more detail and draw the concrete conclusion. Also add comparative studies from literature and draw the conclusion

6.      How the study add contribution to the scientific society? In terms of results and conclusion

Reviewer 2 Report

The paper reports about the TPMS properties dedicated to thermal transpiration pump. Such structures have renewed some interest since their potentials applications in gas dynamics, porous structures , hard resistance …

I have no reservations about the validity of the experiments and results. My remarks concern some minor points

For the average reader, it is interesting to speak about (may be in appendix) the general properties of TPMS and their potential applications. The key points are that deserve to be mentioned are

-hyperbolic geometry (Lynch, M.L.; Spicer, P.T. Bicontinuous Liquid Crystals; CRC Press: Boca Raton, FL, USA, 2005; Volume 127. Meeks, W.H., III. The theory of triply periodic minimal surfaces. Indiana Univ. Math. J. 1990, 39, 877–936.). The hyperbolic geometry plays a role as long heat equation (and then the diffusion) in hyperbolic geometry is quite different from the standard one in Euclidean geometry.

-minimization of the Willmore energy (see Bobenko, A.I. A conformal energy for simplicial surfaces. Comb. Comput. Geom. 2005, 52, 133–143.)

-enhanced mechanical properties (Jin, M., Feng, Q., Fan, X., Luo, Z., Tang, Q., Song, J., ... & Zhao, M. (2022). Investigation on the mechanical properties of TPMS porous structures fabricated by laser powder bed fusion. Journal of Manufacturing Processes, 76, 559-574.

Please mention in the table 1 caption that TPMS equation is a trigonometric  approximation within Weierstrass Enneper formalism

Likewise, a short definition of Poiseuille and Darcy flow will be helpful

The key point is the sentence «  it might be difficult to justify which TPMS structure was better tha another.. » My feeling is that the tortuosity was a key parameter (see Kowalczyk, P.; HoÅ‚yst, R.; Terrones, M.; Terrones, H. Hydrogen storage in nanoporous carbon materials: Myth and facts. Phys.
Chem. Chem. Phys. 2007, 9, 1786–1792., Nicolaï, A.; Monti, J.; Daniels, C.; Meunier, V. Electrolyte Diffusion in Gyroidal Nanoporous Carbon. J. Phys. Chem. C 2015, 119,
2896–2903. See to Nanomaterials 2021, 11, 1694)  

The tortuosity  governs the diffusion and is strongly dependent to the TPMS class

General remark : the physical TPMS differs from the mathematical ones : first of all, the surface thickness, and the periodicity which is not infinite (that introduces boundary effects).

The technical aspects of this work appear sound and the results are sensible. On the other hand, I believe that the paper could be suitable for publication in « Micromachine » after a minor revision

Round 2

Reviewer 1 Report

Paper  can be accepted after including the mesh optimization and selection detail
